# Prognostic and Predictive Significance of Stromal Tumor-Infiltrating Lymphocytes (sTILs) in ER-Positive/HER2−Negative Postmenopausal Breast Cancer Patients

**DOI:** 10.3390/cancers14194844

**Published:** 2022-10-04

**Authors:** Jenny Pousette, Annelie Johansson, Carolin Jönsson, Tommy Fornander, Linda S. Lindström, Hans Olsson, Gizeh Perez-Tenorio

**Affiliations:** 1Department of Biomedical and Clinical Sciences, Linköping University, 581 83 Linköping, Sweden; 2Department of Oncology, Linköping University, 581 83 Linköping, Sweden; 3Department of Oncology and Pathology, Karolinska Institute and University Hospital, 171 64 Stockholm, Sweden; 4Department of Molecular and Immunological Pathology, Linköping University, 581 83 Linköping, Sweden

**Keywords:** immune response, endocrine treatment, anti-tumoral response, T-lymphocytes, B-lymphocytes, immune checkpoint blockade

## Abstract

**Simple Summary:**

The presence of immune cells within the tumor (TILs) indicates good prognosis for some aggressive breast cancer subtypes. However, less is known about TILs role within the hormone-responsive breast cancer (ER+/HER2−). These tumors represent up to 70% of all breast cancers and often exhibit long-term metastasis. TILs were quantified in tumor section samples from 763 postmenopausal women who received tamoxifen vs. no treatment and categorized into low, intermediate, or high. Among the ER+/HER2−, TILs were associated with poor prognostic variables but did not have prognostic value. High TILs indicate less benefit from tamoxifen. Interestingly, high gene expression of some TIL markers did indicate good prognosis even after adjusting for other clinical variables or were associated with less tamoxifen benefit. These results suggest that TIL markers could be used as prognostic, predictive indicators, and potential candidates for immunotherapy interventions after tamoxifen failure.

**Abstract:**

The clinical impact of tumor-infiltrating lymphocytes (TILs) is less known for breast cancer patients with the estrogen receptor-positive (ER+)/human epidermal growth factor receptor-negative (HER−) subtype. Here, we explored the prognostic and predictive value of TILs regarding distant recurrence-free interval (DRFI) and breast cancer-specific survival (BCSS) in 763 postmenopausal patients randomized to receive tamoxifen vs. no systemic treatment. TILs were assessed in whole section tumor samples stained with H&E and divided into low (<10%), intermediate (10–39%), or high (≥40%). High TILs were associated with poor prognostic variables and good prognoses for all patients, but not within the ER+/HER2− group. Within the ER+/HER2− group, high gene expression of CD19 and PD-L1 and high IMMUNE1 score indicated good prognosis in multivariable analysis while high CD8 and CD19 gene expression and high IMMUNE1 score were associated with less tamoxifen benefit. These results indicate that within the ER+/HER2− subtype there could be subsets of patients where expression of specific TIL markers might be used to reveal candidates for immune therapy interventions upon failure of the endocrine therapy.

## 1. Introduction

Breast cancer is a heterogeneous disease that can be classified into different subtypes based on immunohistochemical markers (IHC) [1,2,3] such as the estrogen receptor (ER), progesterone receptor (PR), human epidermal growth factor receptor 2 (HER2), and the proliferation marker Ki67. Similarly, five subgroups with different clinical outcome have been identified by using tumor gene expression information including 50 genes (PAM50) [3,4]. In brief, tumors expressing ER in the absence of HER2 (ER+/HER2−) are often classified as luminal subtype (A or B) [5], tumors with HER2 expression are often of HER2−enriched subtype, whereas tumors with higher proliferation while lacking the three receptors, also denominated triple negative breast cancers (TNBC), are often classified to the basal-like subtype [6]. The largest of these subgroups is the ER+/HER2− comprising up to 70% of breast cancers [7]. Due to its dependency on ER signaling, these tumors are treated with tamoxifen (TAM) or aromatase inhibitors (AI). A canon of ER+ breast cancer is that half or more of the recurrences occur late, up to 20 years or more after diagnosis [8,9]. Therefore, it is important to understand which patients are at risk for late recurrences and how they can be treated upon relapse. In that sense, the immune checkpoint blockade has emerged as a promising therapeutic option for some types of breast cancer [10]. 

Tumor-infiltrating lymphocytes (TILs), comprising natural killer cells (NK), T-, and B-lymphocytes, are defined as infiltration of lymphocytes in the tumor microenvironment [11]. TILs are assessed as the number of mononuclear cells within the tumor that can be present in stroma (sTILs) and intratumoral (iTILs). The interest in TILs has grown in recent decades but its effect and importance in BC is still under investigation [12,13]. Quantification of sTILs is currently under discussion for TNBC [14], where sTILs can be visually assessed throughout the tumor area following international guidelines [15]. High TILs have been observed in HER2+ and TNBC and associated with good prognosis [12,16,17]. However, TNBC and HER2+ tumors are highly mutated and express more tumor-associated antigens, thereby activating a stronger immune response. In contrast, ER+ tumors generally have a lower mutation burden but often carry mutations in the *PIK3CA* gene, which is an important component of the PI3K/AKT survival pathway. There are indications that *PIK3CA*-mutated tumors often attract sTILs, but the clinical significance is not well-established [18,19]. Studies supporting the role of sTILs in ER+ disease are currently emerging [10]. For example, a low amount of sTILs has been associated with benefits from the aromatase inhibitor (AI) exemestane [20] and long-term response to pembrolizumab combined with TAM [21]. These findings suggest that even patients with immunologically “cold” ER+ disease could benefit from immunotherapy combined with endocrine treatments. 

Two lymphocytic populations commonly found in breast cancer are the cytotoxic T-lymphocytes (CTL or CD8+) and the regulatory T-lymphocytes (TREG or CD4+/FOXP3+) [22]. These have been associated with higher risk for recurrence and *PIK3CA* mutations in a subset of ER+/HER2− breast cancers [19] with shorter overall survival [23] and higher risk of recurrence 5 years after diagnosis [24]. This further supports the idea that ER+ breast cancer is immunogenic and that some patients could benefit from immunomodulatory therapies when endocrine therapy fails. Moreover, B-lymphocytes (CD19+) have been associated with improved invasive and overall survival for TNBC and HER2+ BC subtypes and seems to be important for the anti-tumoral immune response [25].

With our study, we want to explore whether sTILs have a prognostic value in patients with ER+/HER2− tumors and to investigate their predictive value for the endocrine treatment with TAM. We also want to identify clinically relevant lymphocytic subpopulations by analyzing TILs in association with previously reported gene expression modules [26,27] as well as individual TIL markers that could potentially guide immunotherapeutic interventions. 

This study represents a unique opportunity to study the prognostic value of sTILs and lymphocytic gene expression markers in patients who did not receive adjuvant endocrine therapy and to reveal its predictive value for patients treated with TAM in a large, randomized study with long-term follow-up.

## 2. Materials and Methods

Randomized clinical trials were conducted in 1780 postmenopausal lymph node-negative breast cancer patients with tumor size less than or equal to 30 mm (STO-3 trial) by the Stockholm Breast Cancer Study Group from 1976 to 1990 [28]. Patients were randomized to receive adjuvant tamoxifen therapy (40 mg daily) or no endocrine therapy (control group or no systemic treatment). In 1983, patients who received tamoxifen therapy without cancer recurrence during the 2-year treatment and who consented to continue participation in the STO-3 study, were further randomized to receive 3 additional years of tamoxifen therapy vs. no endocrine therapy.

All residents in Sweden have a unique national registration number. This number allows automatic linkage with various personal records from national and regional registers, which provides high validity and essentially complete data coverage. Follow-up until 31 December 2016 (complete 25-year follow-up) was available for all patients in the STO-3 trial [29,30]. A flow chart of the patient distribution is shown in Figure 1. 

### 2.1. Definition of Clinical Variables and Tumor Characteristics

Tumor grade was defined according to the Nottingham Histological Grade (NHG) [31]. BC subtypes were determined with immunohistochemistry (IHC) based on the expression of ER, PR, and HER2 receptors [32] or using the PAM50 molecular profile [4]. Based on the IHC, the tumors were classified into ER+/HER2−, HER2+, and TNBC subtypes. ER and PR expression was considered positive when ≥10% of the cell nuclei were stained according to Swedish guidelines. For the proliferation marker Ki67, the cut-off level was set to 15%. HER2 was defined as positive upon strong membrane staining by IHC (IHC3+). The 70-gene signature was used to identify women with ultra-low, low, or high risk of death from breast cancer [33,34].

### 2.2. Immunohistochemical Detection of ER, PR, HER2, and Ki67

ER and PR were detected by immunohistochemistry using CONFIRM™ mouse anti-ER antibody (clone 6F11) and CONFIRM™ mouse anti-PR antibody (clone 16) from Ventana Medical Systems. The staining was performed according to the manufacturer instructions using the Ventana^®^ automated slide stainer (Ventana Medical Systems, S.A., Illkirch, France). HER2 was detected with the DAKO AO0485 polyclonal rabbit antibody also according to the guidelines provided by the manufacturer [35]. 

Ki67 was stained with the monoclonal mouse anti-human Ki67 clone MIB-1 (DAKO M7240) using the DAKO Link48 Auto Stainer protocol [36].

### 2.3. TILs Scoring

TILs were evaluated in 809 whole section formalin-fixed paraffin-embedded (FFPE) tumor tissue samples stained with H&E. The slides were scanned at 200× magnification using a Leica scanner (Leica Aperio CS2, Leica Biosystems, Mount Waverley, VIC, Australia), and the images were visualized using the Aperio ImageScope free software (version 12.3.5.5080).

TILs assessment was independently performed by the senior consultant histopathologist (HO) and one trained researcher (JP). Both were blinded to the clinical data. A consensus decision was made in cases with diverse result.

Briefly, all mononuclear cells within the stroma in the tumor area were considered as sTILs (referred to as TILs) and were estimated as a percentage (1–100%) of the total stromal area within the tumor and its borders. TILs in areas of necrosis, outside of tumor borders, or within in situ carcinomas were excluded. Based on the percentage of TILs, tumor samples were considered to have low (<10%), intermediate (10–39%), or high (≥40%) infiltration. Forty-six samples consisting of in situ carcinoma, microinvasive carcinoma, benign tissue, Paget’s disease, intracystic tumors, or of poor quality preparation were excluded following the exclusion criteria described in the guidelines by the Immuno-Oncology International TILs Working Group (https://www.tilsinbreastcancer.org/) (accessed on 2 February 2021) [37].

### 2.4. Gene Expression Data and Normalization

Gene expression data were independently generated using custom-designed arrays, Agilent Technologies (Santa Clara, CA, USA), containing approximately 32.1K probes, representing approximately 21.5K unique genes from FFPE breast cancer tumor tissue. A total of 652 of 727 breast cancer tumors passed the RNA quality check according to the diagnostic quality model and were used in the analysis. Gene expression data were normalized with quantile normalization: gene expression values were log2-scaled and quantile normalized before analysis [33]. Microdissection was performed before mRNA extraction as a part of the pipeline in the microarray assay. Gene expression levels of CD4, CD8A (hereafter CD8), CD19, FOXP3, programmed cell death protein 1 (PDC1 or PD1), and programmed death ligand-1 (PD-L1) were analyzed using the microarray platform (Agilent Technologies) [33]. The definition of a high expression was set at the median for CD4, CD19, FOXP3, PD-L1, PD1, and the immune gene modules IMMUNE1 and IMMUNE2 [26,27] for all analyses. CD8 was separated in quartiles where CD8 q1–3 = low expression and CD8 q4 = high expression.

### 2.5. Statistical Analysis

TILs-three-group score (low (<10%), intermediate (10–39%), or high (≥40%)) was used for all analyses except for the predictive analysis where <10% and ≥10% TIL groups were used to gain statistical power.

The distribution and correlation of TILs and the previously established clinicopathological variables were analyzed using the Chi^2^ test and Spearman rank-order correlation analysis. 

Two variables were used as endpoints in the survival analysis: distant recurrence-free interval (DRFI) and breast cancer-specific survival (BCSS). DRFI was defined as the period from the date of randomization until detection of distant metastasis, and BCSS was defined as the time from randomization until patient’s death due to breast cancer [38,39]. 

Univariate Kaplan–Meier analysis using DRFI and BCSS as endpoints was performed and the Log-rank test was used to visualize statistical differences in patient survival. Significant difference between groups was also estimated with the Cox proportional hazard modeling for both univariate and multivariable analyses. The following clinical variables were included in the multivariable model: tumor size (<20 mm or ≥20 mm), grade (1–3), ER (negative or positive), PR (negative or positive), HER2 (negative or positive), Ki67 (<15% or ≥15%), and TAM treatment. *p* values were considered significant when *p* < 0.05. 

For statistical analysis, TIBCO Statistica™ version 13.0 was used.

## 3. Results

Based on the material and clinical data from the original STO-3 trial, 809 tumor tissue samples were available in H&E slides. From these, a total of 763 samples were assessed for TILs (Figure 1) and classified as low (<10%) in 604 (79%) cases, intermediate (10–39%) in 69 (9%) cases, and high (≥40%) in 90 (12%) cases. Figure 2 shows representative pictures of the different TILs staining. The tumors were classified into ER+/HER2−, HER2+, and TNBC based on IHC scores. Overall, 458 (73%) tumors were classified as ER+/HER2−, 74 (12%) as HER2+, and 94 (15%) as TNBC (Figure 1).

### 3.1. TILs Association with Clinicopathological Characteristics

Within the ER+/HER2− group, high TILs tended to be associated with larger tumors and significantly associated with higher tumor grade and high Ki67 expression. High and intermediate TILs was often found among the genomic high-risk tumors while the group defined as ultra-low presented only lower TILs. Moreover, tumors with high TILs often presented higher CD8, CD19, PD-L1, and PD1 gene expression levels. Similarly, these tumors were characterized by higher scores of the IMMUNE1 and IMMUNE2 gene expression modules (Table 1).

The same analysis performed in all the patients, independently of the BC subtype, showed similar results to the ER+/HER2− group. Additionally, higher TILs were overrepresented among the TNBC and HER2 subtypes or among basal, luminal B, and HER2−enriched subtypes, according to the PAM50 molecular classification (Appendix A).

### 3.2. TILs, Lymphocytic Markers, and Gene Signatures as a Prognostic Marker

Univariate and multivariable analyses, considering DRFI and BCSS as endpoints, were performed for all patients or for the individual IHC subgroups: ER+/HER2− (Table 2 and Table 3 and Figure 3), HER2+, or TNBC (Appendix A). The univariate analysis, restricted to the control arm, did not reveal a significant prognostic value for TILs regarding DRFI (Table 2 and Appendix A) nor BCSS (Table 3 and Appendix A). 

The multivariable analysis in all patients adjusting for tumor size, tumor grade, ER, PR, HER2, Ki67, and TAM treatment showed that high TILs (≥40%) significantly reduced the risk of distant recurrences with a 53% risk reduction (HR (95% CI) = 0.47 (0.24–0.95)), but not when limiting the analysis to the ER+/HER2− or any other specific IHC subtype (Table 2 and Appendix A). When analyzing prognosis in terms of BCSS, we found that high TILs significantly reduced the risk of breast cancer-related death by 55%, compared with low TILs (HR (95% CI) = 0.45 (0.22–0.93)) in the analysis including all patients, but not when limiting to the ER+/HER2− or any other individual IHC subtype (Table 3 and Appendix A). 

Thus, TILs did not have prognostic value when stratifying for the ER+/HER2− subtype (Table 2 and Table 3 and Figure 3B). 

Table 4 summarizes the univariate Cox regression analysis to estimate the prognostic value of TILs subpopulations based on gene expression of known lymphocytic markers. These results showed that high CD8 expression tended to be associated with good prognosis in the ER+/HER2− subtype HR (95% CI) = 0.45 (0.20–0.98). High CD19 gene expression was significantly associated with a lower risk to develop metastasis in all breast cancer patients: HR (95% CI) = 0.51 (0.33–0.79) and the ER+/HER2− group: HR (95% CI) = 0.26 (0.14–0.51) (Figure 4A,B). Likewise, a higher score of the IMMUNE1 gene module was associated with good prognosis among all subtypes: HR (95% CI) = 0.58 (0.38–0.87), the ER+/HER2− group: HR (95% CI) = 0.49 (0.29–0.84) (Figure 4C,D) and a trend was shown within TNBC: HR (95% CI) 0.25 (0.06–1.02) (Appendix A). Similarly, high PD-L1 expression tended to be associated with good prognosis within all patients: HR (95% CI) = 0.66 (0.43–0.99), ER+/HER2−: HR (95% CI) = 0.60 (0.35–1.03) (Figure 4E,F), and HER2+ subtypes: HR (95% CI) = 0.36 (0.13–1.04) (Appendix A). TILs in groups of low or high expressions of CD4, FOXP3, PD1, and IMMUNE2 showed no significant association with prognosis.

Furthermore, the multivariable Cox regression analysis adjusting for tumor size, grade, Ki67, ER, PR, HER2, and tamoxifen showed that CD19, IMMUNE1, and PD-L1 remained significant prognostic factors among all the patients (Table 5) and for the ER+/HER2− patients (adjusting for tumor size, grade, Ki67, and tamoxifen). Additionally, high IMMUNE1 score tended to have good prognostic value for HER2+ and TNBC subtypes, whereas IMMUNE2 and PD-L1 tended to indicate poor prognosis for TNBC (Appendix A).

### 3.3. Predictive Role of TILs, Lymphocytic Gene Expression Markers, and Gene Signatures for TAM Efficacy

TILs were evaluated as a predictive marker for TAM treatment among patients with ER+/HER2− tumors. Taking DRFI as endpoint for the survival analysis, we found that those patients whose tumors expressed high TILs (≥10%) did not significantly benefit from tamoxifen vs. no TAM (HR (95% CI) = 0.85 (0.21–3.39)), whereas the ones with low TILs (<10%) had significant long-term benefits from TAM (HR (95% CI) = 0.49 (0.31–0.78)). However, the test for interaction neither in univariate (*p* = 0.50) nor multivariable analyses (*p* = 0.67) reached statistical significance (Figure 5 and Table 6). 

Moreover, patients whose tumors expressed high CD8 obtained no significant benefit from tamoxifen vs. no TAM, HR (95% CI) = 1.55 (0.50–4.22), while those with low CD8 expression significantly benefited from TAM, HR (95% CI) = 0.33 (0.20–0.55) (interaction test: *p* = 0.005) (Table 7 and Figure 6A,B). This result remained significant in the multivariable analysis adjusting for tumor size, tumor grade, and Ki67 (Table 7) with *p* value for interaction test, *p* = 0.008.

Similarly, higher CD19 expression indicated no significant benefit from TAM treatment compared with the control group (HR (95% CI) = 1.22 (0.56–2.66)). In contrast, longer DRFI upon TAM treatment was found among patients with low CD19 expression (HR (95% CI) = 0.29 (0.17–0.50)) (test for interaction *p* = 0.003) (Figure 6C,D). Similar results were found in the multivariable analysis: tamoxifen vs. no tamoxifen within low CD19: HR (95% CI) = 0.28 (0.16–0.49) and tamoxifen vs. no tamoxifen within high CD19: HR (95% CI) = 1.21 (0.53–2.76) with *p* value for interaction = 0.003.

Patients with higher score of the immune response module IMMUNE1 showed no significant tamoxifen benefit (HR (95% CI) = 0.67 (0.34–1.33)), whereas patients with low IMMUNE1 score did (HR (95% CI) = 0.35 (0.20–0.61)) with borderline significant interaction test in univariate (*p* = 0.14) and multivariable analyses (*p* = 0.09) (Figure 6E,F).

Gene expression of CD4, FOXP3, PD-L1, PD1, and IMMUNE2 were not found treatment predictive in univariate nor multivariable analyses.

## 4. Discussion

TILs expression was assessed in a large and unique clinical material including low-risk breast cancer patients randomized to receive TAM vs. no systemic treatment and follow-up for 25 years. We found that TILs were associated with other clinical variables usually coupled with bad prognosis. However, in the multivariable analysis, high TILs indicated longer distant recurrence-free interval and breast cancer-specific survival when the analysis included all breast cancer patients. Moreover, high CD19, IMMUNE1, and PD-L1 gene expression levels indicated better prognosis for all breast cancer patients and for the ER+/HER2− group. 

We also found no significant benefit from TAM among patients with ER+/HER2− tumors and high CD8 or CD19 gene expression compared to low, which might suggest that high expression of specific TIL markers could indicate endocrine treatment resistance.

The grade of lymphocytic infiltration was estimated as low (<10%), intermediate (10–39%), or high (≥40%). Most of the patients included in this study presented tumors with low infiltration while only few were scored in the high group. Although previous publications have chosen a higher threshold of ≥50–60% TILs [5,40], we scored higher infiltration as ≥40% TILs to increase the statistical power of the analysis. 

The ER+/HER2− subgroup was characterized by lower TILs compared to the HER2 and TNBC subtypes; however, within this group of relatively less aggressive tumors, high TILs tended to be associated with larger tumor size and was significantly associated with higher tumor grade and high proliferation (≥15% Ki67 expression). Our results agreed with previous publications [5,16,17,40,41].

Although previous reports show that high TILs indicate good prognosis among TNBC and HER2+ groups [5,12,16,17,40,41,42], we could not find that. We believe that the weaker significance in our study compared to others could be due to the lack of inclusion of premenopausal patients and patients with lymph nodal infiltration where the frequency of TNBC and HER2 subtypes used to be higher and, therefore, high TILs could be analyzed within more statistically relevant groups.

High TILs did not have prognostic value within the ER+/HER2− group. Whereas, others have reported that higher levels of TILs within the ER+/HER2− subtype was related to better prognosis [5]. However, there is no consensus in the literature concerning the clinical significance of high TILs for HR+ disease [10]. 

For certain breast cancers, immunotherapy has emerged as an important part of the oncological treatment. For example, for TNBC, immunotherapy has been introduced as a therapy alternative if the tumor has a positive expression of PD-L1. This treatment has been shown to improve patient survival by inhibiting PD-L1 and enhancing the T-cells-modified anti-tumor immune response [43]. In addition, new research suggests that higher TILs could indicate improved response to immunotherapy. Therefore, TILs could be a valuable marker to predict the outcome from immunotherapy [11,44]. This indicates that the assessment of TILs could be relevant in several areas and an important biomarker to include in the clinical practice regarding the prognosis and choice of treatment in BC. 

Regarding TILs as a predictive biomarker for response to TAM treatment in the ER+/HER2− disease, we found that among patients with ER+/HER2− tumors, those with lower but not higher TILs seem to benefit from TAM compared with no treatment. Although these results might suggest that high TILs could be coupled with treatment resistance, the test for interaction did not reach statistical significance. Comparable results were reported where tumors with high TILs (≥50%) did not benefit from TAM in comparison with low TILs (<10%) and intermediate TILs (10–49%) groups [45]. Even though the results were weak, it raises further speculations regarding the benefit of TAM depending on the grade of TILs.

Regarding the previously described mechanism of the immune response to tumor cells, one could ask whether defining different subpopulations of TILs would be relevant to obtain more detailed information about the interaction between the immune system and tumor response. When evaluating different TIL subpopulations based on gene expression data, we found an association between the grade of TILs and positive expression of CD8 (T-cell marker) and CD19 (B-cell marker), respectively. Cytotoxic CD8+ T-cells appear to have an important anti-tumoral activity, as demonstrated in several studies [42,46]. The assessment of CD8 may therefore be a more relevant prognostic marker.

There are diverse results regarding TILs and CD8 expression in the ER+/HER2– subtype. Depending on the clinicopathological features of the tumor, the grade of TILs and its effect seem to differ and appear to be more complex in comparison to the other BC subtypes [47]. For example, Fujimoto et al. present that high TILs are beneficial in ER+/HER2− tumors with high Ki67, indicating improved disease-free survival, but for tumors with low Ki67, high TILs had a negative prognostic impact [48]. Others suggest that high CD8 infiltration in ER+ tumors has unfavorable outcomes [19]. We also found that high CD8 gene expression levels were coupled with poor response to TAM among the ER+ patients with a significant interaction test. Low CD8+ TILs have been associated with benefits from exemestane, suggesting that CD8+ TILs could predict endocrine treatment response [20]. Moreover, TAM treatment itself could induce immunosuppression via TGF-beta-dependent mechanisms with inhibition of the CD8+ function in the tumor microenvironment [49]. Likewise, CD19+ B-TILs can influence the immune response in many ways other than producing antibodies. For example, infiltrating B-lymphocytes can produce immunosuppressive cytokines such as IL10 and TGF beta [25]. Furthermore, B-lymphocytes can also suppress the anti-tumoral response by expression of the PD1 and PD-L1 in solid tumors [50,51,52], which could be reinforced by the effect of tamoxifen triggering PD-L1 expression and concomitant immunoevasion [53]. Interestingly and in line with our results, elevated PD-L1 expression was coupled with increased distant metastasis-free and overall survival in ER+/HER2− breast cancer [54].

Regarding the higher score of the IMMUNE1 gene module, we found it associated with good prognosis in ER+/HER2− subtype and across all breast cancer subtypes and even tended to be associated with less tamoxifen effect. This is in agreement with others showing good prognosis among TNBC and HER2+ [27], but less is known about the predictive role for tamoxifen.

## 5. Conclusions

TILs are overrepresented among the TNBC and HER2 subtypes but cannot be disregarded among ER+/HER2− breast cancers. Within this BC subtype, TILs were correlated with adverse clinical variables, but did not have prognostic or clear predictive value. TILs in all patients, and independently of other clinical variables, indicated longer distant metastasis-free and breast cancer-specific survival. Moreover, TIL markers such as CD19, PD-L1, and the gene module IMMUNE1, all strongly associated with TILs expression, indicated good prognosis while higher CD8, CD19, or IMMUNE1 were associated with less benefit from tamoxifen. Our results encourage further research to identify key TIL markers with prognostic and predictive value in these patients, maybe indicating good candidates for immunotherapy upon tamoxifen failure.

## Figures and Tables

**Figure 1 cancers-14-04844-f001:**
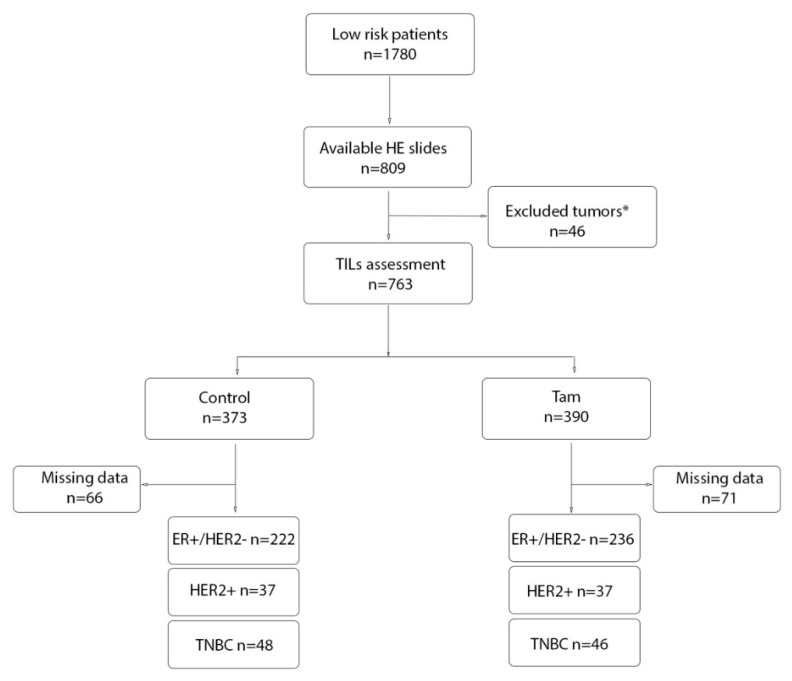
Flow chart of patient distribution. (*) Forty-six tumors were excluded due to being in situ carcinomas, benign, microinvasive, or due to poor quality of the preparate. ER (estrogen receptor), TNBC (triple negative breast cancer), TILs (stromal-infiltrating lymphocytes), HE (Hematoxylin and eosin staining).

**Figure 2 cancers-14-04844-f002:**
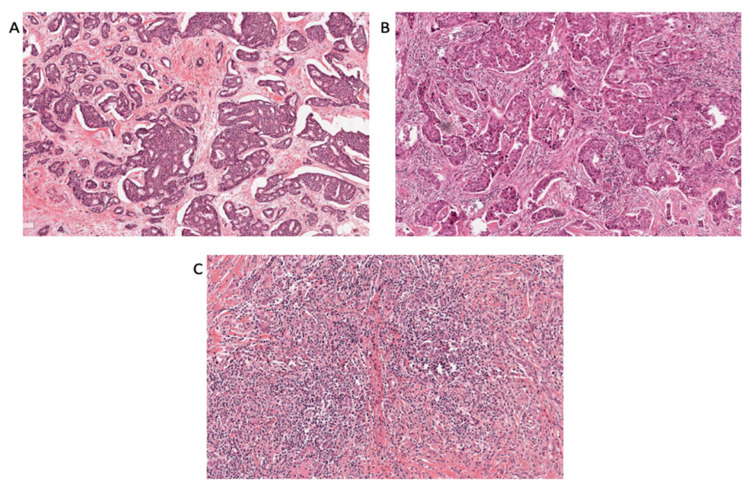
H&E staining of formalin-fixed paraffin embedded whole tumor sections. Representative pictures from a tumor with (**A**) low (<10%), (**B**) intermediate (10–39%), and (**C**) high (≥40%) TILs. The images were taken at 200× in a Leica Aperio CS2 scanner.

**Figure 3 cancers-14-04844-f003:**
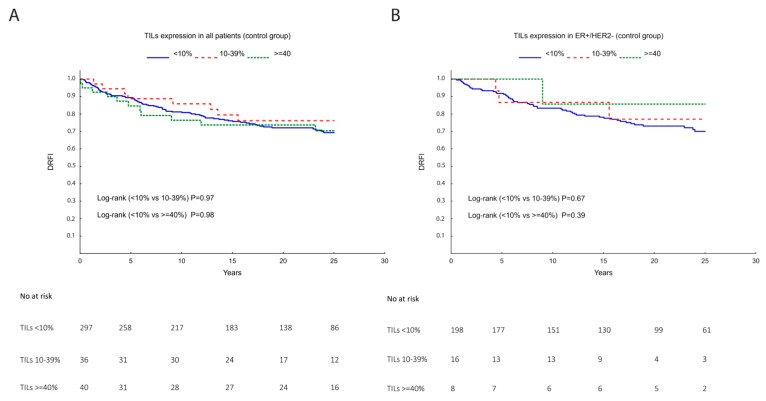
Kaplan–Meier survival analysis showing TILs prognostic value within the control group regarding the endpoint distant recurrence-free interval (DRFI) in (**A**) all patients; (**B**) ER+/HER2−. Tumors were categorized into low (<10%), intermediate (10–39%), or high (≥40%) TILs. Risk tables are shown below the graphs and *p* values correspond to the Log-rank test.

**Figure 4 cancers-14-04844-f004:**
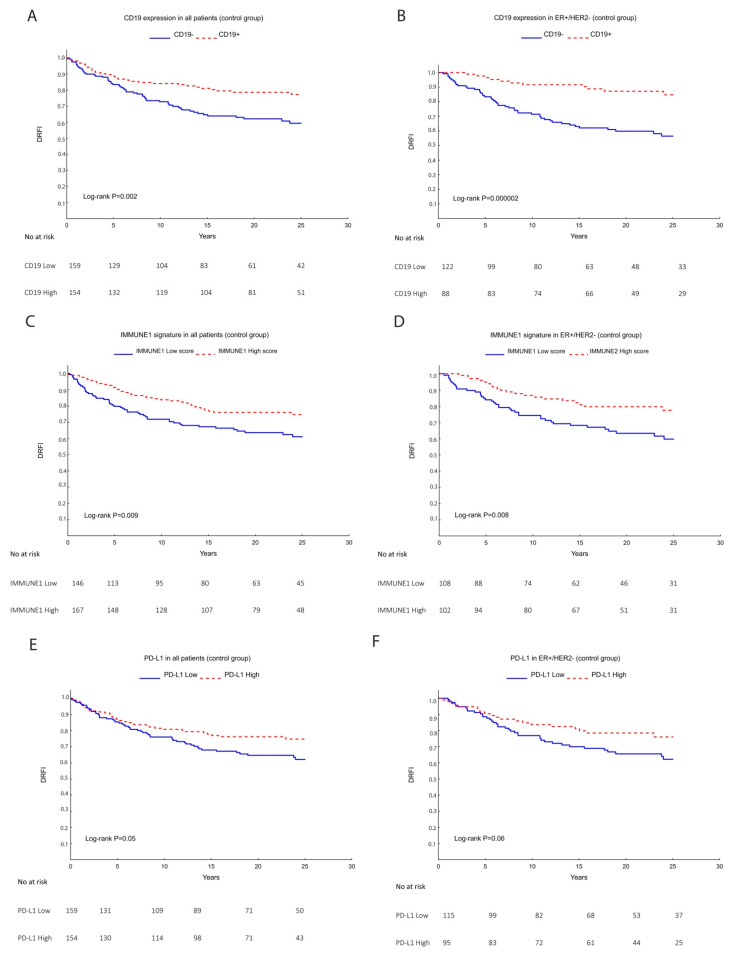
Kaplan–Meier survival analysis within control patients regarding the endpoint distant recurrence-free interval (DRFI). All markers with significant prognostic value after adjusting for multiple clinical variables are shown: CD19 in all the patients (**A**) or in the ER+/HER2− subgroup (**B**), IMMUNE1 signature within all patients (**C**), or within ER+/HER2− subgroup (**D**) and PD-L1 within all patients (**E**), or in the ER+/HER2− group (**F**). *p* values correspond to the Log-rank test and the risk tables are shown below the graphs.

**Figure 5 cancers-14-04844-f005:**
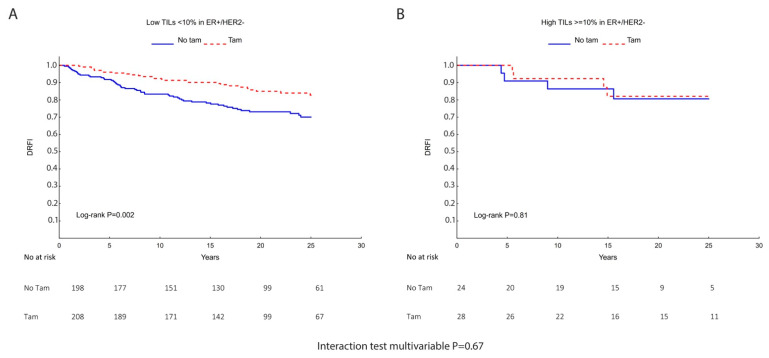
Kaplan–Meier survival analysis showing metastasis-free survival (DRFI) for the subgroup of ER+/HER2− patients randomized to no Tam (control) vs. TAM (Tam). (**A**) Low TILs (<10%); (**B**) high TILs (≥10%). The Log-rank *p* value is inserted within the graphs and below the *p* value from the multivariable interaction test between TILs and TAM as well as the risk tables.

**Figure 6 cancers-14-04844-f006:**
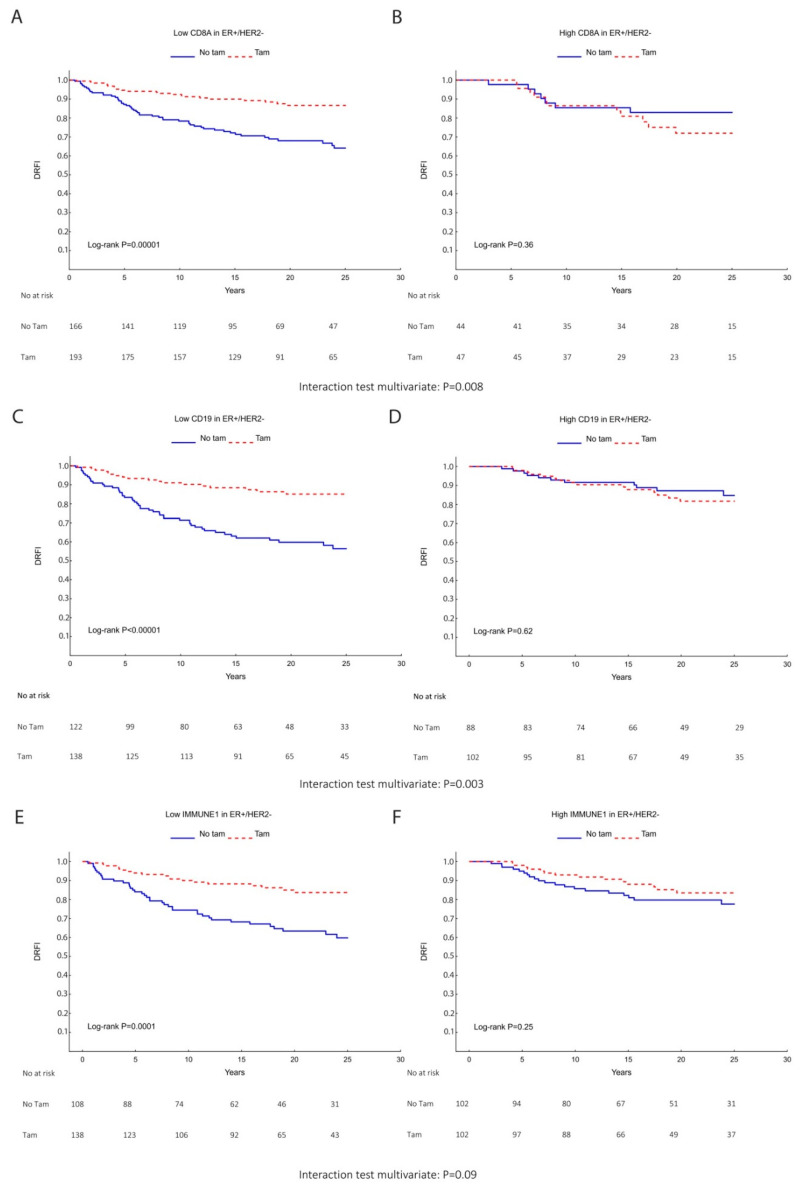
Kaplan–Meier survival analysis with *p* values from the Log-rank test showing differences in distant recurrence-free interval (DRFI) for the subgroup of ER+/HER2− patients allocated to control (No Tam) vs. TAM (Tam). (**A**) Low CD8 expression (Q1–3q1–3) and (**B**) high CD8 expression (q4), (**C**) low CD19, (**D**) high CD19, (**E**) low IMMUNE1 score, and (**F**) high IMMUNE1 score. Risk tables and *p* values from the multivariable interaction test between TIL markers and TAM are indicated below.

**Table 1 cancers-14-04844-t001:** TIL scores in association with clinicopathological variables within the ER+/HER2− subgroup.

Variables	Low TILs <10% n (%)	Intermediate TILs 10–39% n (%)	High TILs ≥40% n (%)	*p* Value
Tumor size	N = 450			0.07
≤20 mm	337 (90)	24 (6)	14 (4)	
>20 mm	62 (83)	8 (10)	5 (7)	
Tumor grade (NHG)	N = 394			**0.001**
1	83 (95)	4 (5)	0 (0)	
2	223 (88)	16 (6)	14 (6)	
3	42 (78)	7 (13)	5 (9)	
Ki67	N = 355			**<0.0001**
<15%	252 (92)	17 (6)	5 (2)	
≥15%	60 (74)	8 (10)	13 (16)	
70-gene signature	N = 351			**<0.0001**
Ultra-low risk	65 (100)	0 (0)	0 (0)	
Low risk	163 (90)	13 (7)	5 (3)	
High risk	82 (78)	11 (11)	12 (11)	
CD4	N = 351			0.99
Low	145 (88)	10 (6)	9 (5)	
High	165 (88)	14 (7)	8 (4)	
CD8A	N = 351			**<0.0001**
Low	176 (96)	8 (4)	0 (0)	
High	134 (80)	16 (10)	17 (10)	
CD19	N = 351			**<0.0001**
Low	199 (96)	6 (3)	1 (1)	
High	111 (77)	18 (12)	16 (11)	
FOXP3	N = 351			0.45
Low	155 (90)	11 (6)	7 (4)	
High	155 (87)	13 (7)	10 (6)	
PD-L1	N = 351			**0.005**
Low	181 (92)	13 (7)	2 (1)	
High	129 (83)	11 (7)	15 (10)	
PD1	N = 351			**<0.0001**
Low	194 (97)	3 (2)	2 (1)	
High	116 (76)	21 (14)	15 (10)	
IMMUNE1	N = 351			**<0.0001**
Low	178 (96)	4 (2)	3 (2)	
High	132 (80)	20 (12)	14 (8)	
IMMUNE2	N = 351			**<0.0001**
Low	191 (98)	2 (1)	2 (1)	
High	119 (76)	22 (14)	15 (10)	

In bold, significant *p* values < 0.05.

**Table 2 cancers-14-04844-t002:** Univariate and multivariable Cox regression analyses of DRFI as an endpoint to assess the prognostic value of TILs in all patients and the ER+/HER2− breast cancer subtype.

TILs	Univariate DRFI ^a^ HR (95% CI)	Multivariable DRFI ^b^ HR (95% CI)
All patients	(n = 373)	(n = 456)
<10% (Ref)	1.00	1.00
10–39%	0.77 (0.37–1.60)	0.85 (0.45–1.60)
≥40%	1.01 (0.54–1.90)	**0.47 (0.24–0.95)**
ER+/HER2−	(n = 222)	(n = 348)
<10% (Ref)	1.00	1.00
10–39%	0.77 (0.24–2.46)	0.99 (0.39–2.47)
≥40%	0.44 (0.06–3.28)	0.37 (0.09–1.64)

^a^ Univariate analysis including untreated patients (control). ^b^ Multivariable analysis adjusting for tumor size, tumor grade, Ki67, ER, PR, HER2, and TAM (all subtypes) or adjusted for tumor size, tumor grade, Ki67, and TAM (ER+/HER2−). (Ref) Reference in bold, statistically significant Hazard Ratios.

**Table 3 cancers-14-04844-t003:** Univariate and multivariable Cox regression analyses of BCSS as an endpoint to assess the prognostic value of TILs in all patients and the ER+/HER2− breast cancer subtype.

TILs	Univariate BCSS ^a^ HR (95% CI)	Multivariable BCSS ^b^ HR (95% CI)
All patients	(n = 373)	(n = 456)
<10% (ref)	1.00	1.00
10–39%	0.82 (0.37–1.78)	0.84 (0.43–1.64)
≥40%	1.09 (0.56–2.11)	**0.45 (0.22–0.93)**
ER+/HER2−	(n = 222)	(n = 348)
<10% (ref)	1.00	1.00
10–39%	0.94 (0.29–3.02)	1.09 (0.43–2.76)
≥40%	0.54 (0.07–3.95)	0.42 (0.10–1.92)

^a^ Univariate analysis including untreated patients (control). ^b^ Multivariable analysis adjusting for tumor size, tumor grade, Ki67, ER, PR, HER2, and TAM (all subtypes) or adjusted for tumor size, tumor grade, Ki67, and TAM (ER+/HER2−). (Ref) Reference in bold, statistically significant Hazard Ratios.

**Table 4 cancers-14-04844-t004:** Univariate Cox regression analysis of DRFI as an endpoint to assess the prognostic value of different TIL markers and two immune gene modules in untreated patients.

	CD8 ^a^	CD4	FOXP3	CD19	IMMUNE1	IMMUNE2	PD1	PD-L1
Univariate DRFI HR (95% CI)								
All patients (n = 313)								
Low	1.00	1.00	1.00	1.00	1.00	1.00	1.00	1.00
High	0.74 (0.44–1.22)	0.92 (0.61–1.40)	0.74 (0.49–1.12)	**0.51** (**0.33**–**0.79**)	**0.58** (**0.38**–**0.87**)	1.21 (0.80–1.83)	1.21 (0.80–1.83)	**0.66** (**0.43**–**0.99**)
ER+/HER2− (n = 210)								
Low	1.00	1.00	1.00	1.00	1.00	1.00	1.00	1.00
High	**0.45** (**0.20**–**0.98**)	0.85 (0.51–1.41)	0.67 (0.40–1.13)	**0.26** (**0.14**–**0.51**)	**0.49** (**0.29**–**0.84**)	0.87 (0.52–1.47)	1.36 (0.81–2.26)	**0.60** (**0.35**–**1.03**)

^a^ CD8 Low = q1–3, CD8 High = q4 otherwise the median was used as cut-off. In bold, statistically significant, or borderline Hazard Ratios.

**Table 5 cancers-14-04844-t005:** Multivariable Cox regression analysis of DRFI as an endpoint to assess the prognostic value of different TIL markers and two immune gene modules.

	CD8 ^a^	CD4	FOXP3	CD19	IMMUNE1	IMMUNE2	PD1	PD-L1
Multivariable DRFI ^b^ HR (95% CI)								
All patients (n = 524)								
Low	1.00	1.00	1.00	1.00	1.00	1.00	1.00	1.00
High	0.79 (0.52–1.20)	0.83 (0.58–1.19)	0.84 (0.58–1.19)	**0.63** (**0.43**–**0.93**)	**0.57** (**0.39**–**0.83**)	1.11 (0.75–1.64)	1.36 (0.92–2.01)	**0.60** (**0.41**–**0.89**)
ER+/HER2− (n = 418)								
Low	1.00	1.00	1.00	1.00	1.00	1.00	1.00	1.00
High	0.71 (0.41–1.24)	0.79 (0.51–1.21)	0.84 (0.55–1.29)	**0.46** (**0.29**–**0.74**)	**0.63** (**0.41**–**0.98**)	0.88 (0.57–1.37)	1.46 (0.95–2.26)	**0.54** (**0.34**–**0.85**)

^a^ CD8 Low = q1–3, CD8 High = q4 otherwise the median was used as cut-off. ^b^ Multivariable analysis adjusting for tumor size, tumor grade, Ki67, ER, PR, HER2, and TAM (all subtypes) or adjusted for tumor size, tumor grade, Ki67, and TAM (ER+/HER2−). In bold, statistically significative, or borderline Hazard Ratios.

**Table 6 cancers-14-04844-t006:** Univariate and multivariable analyses to assess TILs predictive value for TAM efficacy vs. no endocrine treatment within ER+/HER2− patients.

TILs	N	Tamoxifen	Univariate DRFI HR (95% CI)	Interaction	N	Tamoxifen	Multivariable DRFI ^a^ HR (95% CI)	Interaction
Low <10%	406	−	1.00		306	−	1.00	
		+	0.49 (0.31–0.78)			+	0.46 (0.27–0.79)	
				*p* = 0.50				*p* = 0.67
High ≥10%	52	−	1.00		42	−	1.00	
		+	0.85 (0.21–3.39)			+	0.38 (0.06–2.28)	

^a^ Multivariable analysis adjusting for tumor size, tumor grade, and Ki67.

**Table 7 cancers-14-04844-t007:** Predictive value for TAM efficacy within the ER+/HER2− patients of TIL markers and two immune gene modules.

TIL Markers	N	Tamoxifen	Univariate DRFI HR (95% CI)	Interaction	N	Tamoxifen	Multivariable DRFI ^a^ HR (95% CI)	Interaction
CD8 Low	359	−	1.00		333	−	1.00	
		+	**0.33** (**0.20**–**0.55**)			+	**0.31** (**0.19**–**0.53**)	
				***p* = 0.005**				***p* = 0.008**
CD8 High	91	−	1.00		84	−	1.00	
		+	1.55 (0.60–4.02)			+	1.31 (0.48–3.60)	
CD4 Low	226	−	1.00		214	−	1.00	
		+	0.53 (0.30–0.92)			+	0.50 (0.28–0.9)	
				*p* = 0.37				*p* = 0.42
CD4 High	224	−	1.00		84	−	1.00	
		+	0.35 (0.18–0.70)			+	0.33 (0.16–0.71)	
FOXP3 Low	227	−	1.00		209	−	1.00	
		+	0.34 (0.19–0.62)			+	0.35 (0.18–0.65)	
				*p* = 0.14				*p* = 0.25
FOXP3 High	223	−	1.00		209	−	1.00	
		+	0.64 (0.35–1.18)			+	0.56 (0.30–1.05)	
CD19 Low	260	−	1.00		242	−	1.00	
		+	**0.29** (**0.17**–**0.50**)			+	**0.28** (**0.16**–**0.49**)	
				***p* = 0.003**				***p* = 0.003**
CD19 High	190	−	1.00		176	−	1.00	
		+	1.22 (0.56–2.66)			+	1.21 (0.53–2.76)	
PD1 Low	257	−	1.00		242	−	1.00	
		+	0.46 (0.25–0.85)			+	0.40 (0.21–0.76)	
				*p* = 0.92				*p* = 0.77
PD1 High	193	−	1.00		176	−	1.00	
		+	0.44 (0.24–0.81)			+	0.42 (0.22–0.78)	
PD–L1 Low	252	−	1.00		240	−	1.00	
		+	0.42 (0.25–0.71)			+	0.37 (0.22–0.64)	
				*p* = 0.61				*p* = 0.45
PD–L1 High	198	−	1.00		178	−	1.00	
		+	0.53 (0.26–1.08)			+	0.52 (0.24–1.13)	
IMMUNE1 Low	246	−	1.00		228	−	1.00	
		+	**0.35** (**0.20**–**0.61**)			+	**0.31** (**0.18**–**0.56**)	
				*p* = 0.14				***p* = 0.09**
IMMUNE1 High	204	−	1.00		190	−	1.00	
		+	0.67 (0.34–1.33)			+	0.68 (0.33–1.39)	
IMMUNE2 Low	259	−	1.00		240	−	1.00	
		+	0.38 (0.21–0.68)			+	0.32 (0.17–0.59)	
				*p* = 0.31				*p* = 0.14
IMMUNE2 High	191	−	1.00		178	−	1.00	
		+	0.59 (0.31–1.10)			+	0.62 (0.33–1.18)	

^a^ Multivariable analysis adjusting for tumor size, tumor grade, and Ki67. In bold, statistically significant Hazard Ratios and *p* values.

## Data Availability

The patient data included in this article cannot be shared publicly due to patient privacy.

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
