# Peer review of "Prognostic and Predictive Significance of Stromal Tumor-Infiltrating Lymphocytes (sTILs) in ER-Positive/HER2−Negative Postmenopausal Breast Cancer Patients"

_cancers, 2022, doi:10.3390/cancers14194844_

Round 1
Reviewer 1 Report
The presence of tumor-infiltrating lymphocytes (TILs) reflects an adaptive anti-tumor immune response. TILs are emerging as biomarkers for predicting clinical response to chemotherapy for breast cancer. In recent translational studies from large prospective trials (e.g., the BIG2–98 trial), it was found that high TIL levels correlate with improved clinical outcome in triple-negative breast cancer (TNBC). However, less is known about TILs in the estrogen receptor ER+/HER2- breast cancer subtype.
Here, Pousette et al. addressed this question by exploring the prognostic and predictive value of TILs -regarding DRFI and BCSS- in postmenopausal patients who received tamoxifen or not.
Significance: If a positive correlation could be found between TILs/DRFI/BCSS, further studies testing TIL levels as a biomarker that predicts clinical benefit for adjuvant chemotherapy in ER-positive/HER2-negative cancer would be warranted.
General comments:
Recently, several articles came out about Tumor-infiltrating lymphocytes as a prognostic and tamoxifen predictive marker in breast cancer. The only “novel” aspect of this work would be the fact that this present study was made on post-menopausal patients, while others were done in pre-menopausal patients. However, as the authors themselves mentioned, the exclusion of pre-menopausal patients and patients with lymph node infiltration might affect the significance of the correlation. Hence, the authors are strongly encouraged to specify that in the title.
Specific comments:
· The authors say patients in control group received no endocrine therapy (page 3, line 4). However, they do not mention about other possible therapies. The authors should specific if the patients were treatment-naïve and, if not, what therapy they received. In this case it would be recommended to further stratify the control group in sub-classes according to such treatment.
· The authors investigated the gene expression level of CD4, CD8, CD19, FOXP3 PD1 and PDL1. However, these data at gene expression level should be validated at the protein level too, for example by IHC on the same tissue slides used for TILs scoring.
· The authors should include how they normalized the expression of CD4, CD8, CD19, FOXP3 PD1 and PDL1.
· How can the authors exclude the contribution of stromal/ necrosis areas in the microarray assay? Did the authors perform manual microdissection before RNA extraction from the sections?
· The authors should provide the specific clone/catalog number/brand of the antibodies used in IHC for ER/PR/HER2/Ki67 and at which dilution they were used.
· Did the authors apply the Kolmogorov-Smirnov test to test for the normal distribution of continuous variables, including TIL levels?
· The authors should look for the composition of immune cells beyond TIL count in the population to rule out potential effects (in terms of TAM treatment) not related to TIL compartment.
Author Response
Dear reviewer, thank you for your valuable comments intended to improve the quality of our work. Please, find the detailed answers to your questions below.
Looking forward your response,
Sincerely yours,
/Gizeh Perez-Tenorio
Corresponding author on behalf of the authors
General comments:
Q1. Recently, several articles came out about Tumor-infiltrating lymphocytes as a prognostic and tamoxifen predictive marker in breast cancer. The only “novel” aspect of this work would be the fact that this present study was made on post-menopausal patients, while others were done in pre-menopausal patients. However, as the authors themselves mentioned, the exclusion of pre-menopausal patients and patients with lymph node infiltration might affect the significance of the correlation. Hence, the authors are strongly encouraged to specify that in the title.
A1. After considering your comment we have now specified in the title that the study was conducted in postmenopausal breast cancer patients. Regarding the novel aspects of our work, we would like to indicate that this study is unique due to 1)-the long follow-up time (>25 years), 2)-the fact that we can study the prognostic impact of TILs in a non-systemic treated group of patients and 3)-the possibility to count with a well clinically characterized group of postmenopausal patients. This analysis would otherwise have been impossible due to the established predictive effect of tamoxifen as proved in 1990 by the Oxford overview of adjuvant trials.
Specific comments:
Q2. The authors say patients in control group received no endocrine therapy (page 3, line 4). However, they do not mention about other possible therapies. The authors should specific if the patients were treatment-naïve and, if not, what therapy they received. In this case it would be recommended to further stratify the control group in sub-classes according to such treatment.
A2. The Stockholm-3 (STO-3) clinical trial started to recruit postmenopausal patients in 1976 to evaluate the effect of tamoxifen in the adjuvant setting in a variety of breast cancer patients. At the time the effect of tamoxifen in estrogen-receptor positive patients was not established and therefore all the breast cancer subtypes were represented within the included patient population. Furthermore, in the mid 70’s the effect of the drug was not well known among premenopausal patients and only postmenopausal patients were included. Although the STO-3 trial included both high risk (node-positive) and low risk (node-negative) patients, in our study we have only included the low-risk group of patients defined as women without lymph nodal infiltration and a tumor diameter-measured on the surgical specimen of less than 30 mm. Most of these patients were only treated with a modified radical mastectomy (n=1348) and few of them (n=432) were treated with conservative breast surgery and radiotherapy (RT) (50 Gy/5 weeks) to the breast parenchyma. These patients were thereafter randomized to receive tamoxifen or no other systemic treatment. Since the DRFI and BCSS after comparing the control group who went through breast conserving surgery + RT and the group treated with radical mastectomy was the same, we considered these groups homogeneous and didn’t subdivide the control group. We have now specified in the text that the control group was equivalent to no systemic treatment (see page 3, line 4)
Q3. The authors investigated the gene expression level of CD4, CD8, CD19, FOXP3, PD1 and PDL1. However, these data at gene expression level should be validated at the protein level too, for example by IHC on the same tissue slides used for TILs scoring.
A3. We do agree on the importance of protein validation. However, we couldn’t perform the IHC analysis on the same tissue slides used for TILs scoring due to lack of whole sections from the same tumors. We count with TMA slides but only for a small fraction of these patients. However, recent reports indicate that gene expression analyses of several genes such as, CD8 and PD-L1 can be as reliable as immunohistochemical analyses (https://doi.org/10.1046/j.1464-410X.2002.02993.x) and
https://doi.org/10.1002/1878-0261.12654) therefore this technique is becoming a more accepted tool for identifying suitable immunotherapy candidates (https://doi.org/10.18632/oncotarget.13691).
Q4 The authors should include how they normalized the expression of CD4, CD8, CD19, FOXP3 PD1 and PDL1.
A4. Yes, a new section 2.4. “Gene expression data and normalization” was added on page 4 as well as a reference (#33). Briefly, “Agilent microarray profiling was performed in 2014. Gene expression data were independently generated using custom-designed arrays, Agilent Technologies (CA, USA), containing approximately 32.1K probes, representing approximately 21.5K unique genes from FFPE breast cancer tumor tissue. 652 of 727 breast cancer tumors passed the RNA quality check according to the diagnostic quality model and were used in the analysis. Gene expression data was normalized with quantile normalization: gene expression values were log2-scaled and quantile normalized before analysis”.
Q5. How can the authors exclude the contribution of stromal/ necrosis areas in the microarray assay? Did the authors perform manual microdissection before RNA extraction from the sections?
A5. Yes, we have now added this information under 2.4. “Gene expression data and normalization, on page 4: “microdisection was performed before mRNA extraction as a part of the pipeline in the microarray assay”.
Q6. The authors should provide the specific clone/catalog number/brand of the antibodies used in IHC for ER/PR/HER2/Ki67 and at which dilution they were used.
A6. This information has been provided on page 3 under “2.2. Immunohistochemical detection of ER, PR, HER-2 and Ki67”. Shortly, “ER and PR were detected by immunohistochemistry using Ventana® automated slide stainer (Ventana Medical Systems, S.A., Illkirch, France) using CONFIRM™ mouse anti-ER antibody (clone 6F11) and CONFIRM™ mouse anti-PR antibody (clone 16) from Ventana Medical Systems. HER2 was detected with the DAKO AO0485 polyclonal rabbit antibody according to the guidelines provided by the manufacturer. Ki67 was stained with the monoclonal mouse anti-human Ki-67 clone MIB-1 (DAKO M7240) using the DAKO Link48 Auto Stainer protocol”l. Moreover, two references (#35 and #36) have been added to the text.
Q7. Did the authors apply the Kolmogorov-Smirnov test to test for the normal distribution of continuous variables, including TIL levels?
A7. In the case of TILs we used three categories to stratify the tumors into low TILs (<10%), intermedium TILs (10-39%) and high TILs ((≥40%). Other variables such as gene expression of the different lymphocytic markers were dichotomous. Thus, we did not apply the Kolmogorov-Smirnov to test for normal distribution of continuous variables since we didn’t use continuous variables in the statistical analysis included.
Q8. The authors should look for the composition of immune cells beyond TIL count in the population to rule out potential effects (in terms of TAM treatment) not related to TIL compartment.
A8. We agree. This could be an interesting follow-up of the present study: to look at the expression of infiltrating macrophages, dendritic cells, neutrophils, myeloid-derived suppressor cells (MDSC) and other cells that might have an impact on the effect of tamoxifen. However, given the scarcity of the present material we only focused on TILs counting and gene expression analyses.
Reviewer 2 Report
An interesting manuscript in which the authors analyzed a significant number of patients in terms of many factors that may have a prognostic value. The main aim of the study was to assess the presence of TIL as a prognostic factor. The manuscript is well designed and statistically compiled and is amenable to publication as is.
Author Response
Dear reviewer, thank you for your comments. We really appreciate that you found our results interesting and the article suitable for publication as it was.
Sincerely yours,
Gizeh Perez Tenorio
Corresponding author on behalf of the other authors
Reviewer 3 Report
The clinical impact of tumor-infiltrating lymphocytes (TILs) is less known for breast cancer patients with the estrogen receptor positive (ER+)/human epidermal growth factor receptor (HER-) subtype. In this article, the authors examined the predictive value of TIL for long-term disease-free interval (DRFI) and specific breast cancer survival (BCSS) in 763 postmenopausal patients randomized to receive tamoxifen compared with no systemic treatment. High TILs were associated with poor prognostic variables and good prognosis in all patients, but not in the ER+/HER2- group.
1. The authors put in the title of the article that they consider only the ER + / HER2- subtype of breast cancer, however, other subtypes are also discussed in the text of the article, I think this is superfluous. If the authors want to keep this material in the article, then it is necessary to change the title and redo the abstract and introduction. If you leave this information, then the emphasis is placed in the article incorrectly, you need to process it.
2. How would you explain the fact that in univariate analysis, high TILs do not affect survival rates, while in multivariate analysis they significantly reduce mortality rates (Tables 2, 3)? Why do univariate and multivariate analyzes have different numbers of patients? How then to compare the results?
3. The drawings are very small and of poor quality, hard to read.
Author Response
Dear reviewer, we really appreciated that you took the time to revise our manuscript. Please find the detailed answers to your questions below.
Looking forward your response,
Sincerely yours
Gizeh Perez-Tenorio,
Corresponding author on behalf of the other authors
Q1. The authors put in the title of the article that they consider only the ER + / HER2- subtype of breast cancer, however, other subtypes are also discussed in the text of the article, I think this is superfluous. If the authors want to keep this material in the article, then it is necessary to change the title and redo the abstract and introduction. If you leave this information, then the emphasis is placed in the article incorrectly, you need to process it.
A1. Thank you for your comment. We included the HER2+ and TNBC subtypes since we believe it would be interesting to see the impact of TILs within these subtypes in this material. However, we do agree that the title and main results are leading the attention toward the ER+/HER2- subtype. Therefore, after careful consideration we kept the ER+/HER2- subgroup in the title and instead moved the information concerning the HER2+ and TNBC subtypes to supplements (Supplementary Tables S2-S5 and Supplementary Figure S1). More specifically:
Table 2 (page 8) and Table 3 (page 9) were modified and the information regarding HER2+ and TNBC was transferred to Supplementary Table S2 and Supplementary Table S3.
Figure 3 (page 10) was modified and the information regarding HER2+ and TNBC was transferred to Supplementary Figure S1.
Table 4 (page 10-11) was modified and the information regarding HER2+ and TNBC was transferred to Supplementary Table S4.
Table 5 (page 13) was modified and the information regarding HER2+ and TNBC was transferred to Supplementary Table S5.
We hope that these changes contribute to redirect the emphasis of the paper.
Q2. How would you explain the fact that in univariate analysis, high TILs do not affect survival rates, while in multivariate analysis they significantly reduce mortality rates (Tables 2, 3)?
A2. Univariate and multivariable analysis differ. Univariate analysis (for example, Kaplan Meier plots with Log-rank test) only considers one variable (TILs in our case) affecting the patient’s survival without metastases (DRFI) or from the disease (BCSS). We performed our univariate analysis in the untreated patients since we had the possibility to study the truly prognostic effect of TILs in absence of any systemic treatment (tamoxifen). Univariate analysis is not as robust as multivariable analysis where the joint effect of several variables is incorporated into the model. As indicated below Table 2 and 3, in the multivariable analysis, we analyzed the prognostic effect of TILs adjusting for tumor size, tumor grade, Ki67, ER, PR, HER2, and treatment (TAM). All these clinical variables (except treatment) were found correlated with TILs and by incorporating them into the model we made sure that the prognostic effect that we observed was due to the TILs contribution and not to any other confounding factor. Multivariable analyses often weight more to prior to taking clinical decisions.
Q2.1Why do univariate and multivariate analyzes have different numbers of patients? How then to compare the results?
A2.1 Univariate and multivariable analyses have different number of patients because the first one only analyzed the prognostic value of TILs in the control arm (systemically untreated patients). However, in the multivariable analysis we choose to include the treatment (tamoxifen) in the model to compensate for any possible effect of the drug. Therefore, we see that this analysis has more patients than the univariate one. The results are telling us that TILs is an independent prognostic factor after compensating for the aforementioned clinical variables which were associated with TILs.
Q3. The drawings are very small and of poor quality, hard to read.
A3. We have now improved the picture resolution to 600 dpi as well as enlarged the size of the text.
Round 2
Reviewer 3 Report
I have no more comments on the article. I believe that in its present form the article can be recommended for publication.